# Dielectrophoretic Separation of Particles Using Microfluidic Chip with Composite Three-Dimensional Electrode

**DOI:** 10.3390/mi11070700

**Published:** 2020-07-20

**Authors:** Li Chen, Xing Liu, Xiaolin Zheng, Xiaoling Zhang, Jun Yang, Tian Tian, Yanjian Liao

**Affiliations:** Key Laboratory of Biorheological Science and Technology, Chongqing University, Ministry of Education, Bioengineering College, Chongqing University, Chongqing 400030, China; biochenli@cqu.edu.cn (L.C.); 201819131055@cqu.edu.cn (X.L.); zxl@cqu.edu.cn (X.Z.); zhangxiaoling@cqu.edu.cn (X.Z.); bioyangjun@cqu.edu.cn (J.Y.)

**Keywords:** microfluidic chip, dielectrophoresis, particle separation, microelectrode, composite

## Abstract

Integrating three-dimensional (3D) microelectrodes on microfluidic chips based on polydimethylsiloxane (PDMS) has been a challenge. This paper introduces a composite 3D electrode composed of Ag powder (particle size of 10 nm) and PDMS. Ethyl acetate is added as an auxiliary dispersant during the compounding process. A micromachining technique for processing 3D microelectrodes of any shape and size was developed to allow the electrodes to be firmly bonded to the PDMS chip. Through theoretical calculations, numerical simulations, and experimental verification, the role of the composite 3D microelectrodes in separating polystyrene particles of three different sizes via dielectrophoresis was systematically studied. This microfluidic device separated 20-, 10-, and 5-μm polystyrene particles nondestructively, efficiently, and accurately.

## 1. Introduction

With microfluidic technology, operations such as sample preparation [1], biological and chemical reactions [2], and separation and detection can be completed within one chip only a few square centimeters in size [3]. Compared with traditional technologies, microfluidic technology has several advantages, such as a low cost, low sample consumption, and rapid [4] and high-sensitivity detection [5]. It exhibits considerable potential for many applications, e.g., biomedicine [6], drug synthesis and screening [7], environmental detection [8], health quarantines [9], forensic identification, and biological reagent detection [10]. Cell and particle separation is another important application of microfluidic technology. Cell separation can be achieved via electrokinetic, magnetic, and hydrodynamic forces [11,12], among others.

Electrokinetic technologies, such as electrophoresis and dielectrophoresis [13,14], are widely used to separate cells, particles, and other types of biological samples [15]. The separation is induced by loading a peripheral electrical signal with integrated microelectrodes on a microfluidic device. Additionally, these microelectrodes can be used as sensors to measure the temperature [16], pH [17], and impedance of the solution, for example. By monitoring the environmental variation, the chemical reaction process [18] can be determined. Two-dimensional (2D) planar metal electrodes are widely used in electrokinetic particles or cell separation because of the relatively mature fabrication techniques, e.g., liftoff and etching. However, these fabrication processes require evaporating or sputtering the metal film on a substrate as an electrode material, necessitating expensive equipment and complex operations. Other planar electrodes, such as ITO electrode [19] and printed carbon electrode, required complex micromachining processes. Furthermore, the vertical distribution of the electric field induced by the planar electrodes is non-uniform. The electric-field intensity is exponentially attenuated with the increasing vertical distance from the electrode [20].

To overcome the foregoing limitations of planar electrodes, various three-dimensional (3D) microelectrodes have been developed for cell and particle manipulation [21]. In contrast to planar microelectrodes, the electric-field distribution in the vertical direction is uniform, which prevents undesired cell or particle movement in vertical direction and disrupts precise manipulation. Additionally, 3D electrodes can withstand larger currents and have lower heat generation than planar electrodes at the same current intensity, which is critical for the survival of cells and other small biological particles [22]. Because of these advantages of 3D microelectrodes, diverse 3D microelectrodes have been developed and integrated on microfluidics chips, such as Si 3D microelectrodes, Au 3D microelectrodes, and 3D carbon microelectrodes. A thin-film 3D metal electrode, which was fabricated by sputtering an Au film on a polyimide microchannel sidewall, was designed to integrate 3D microelectrodes on a microfluidic device by Hu et al. [23]. However, these types of microelectrodes require expensive and complex fabrication processes, including low-pressure chemical vapor deposition, inductively coupled plasma, plasma-enhanced chemical vapor deposition, and electroplating, which limits the application of microfluidic devices [24]. Thus, an inexpensive and highly adaptable microfluidic-chip material of microelectrodes and a matching facile fabrication process would have considerable application potential for electrokinetic manipulation.

As a polymer material, polydimethylsiloxane (PDMS) had been widely used to fabricate microfluidic chips owing to its facile soft lithography-based fabrication process, high optical transmission, and good biocompatibility [25]. However, pure PDMS cannot be used as an electrode, because it is a nonconductive material and does not easily bond with metal films. To ensure that PDMS played the roles of a microfluidic channel and a microelectrode, a mixture of PDMS and conductive micro/nanoparticles was developed and tested. Rao et al. filled a PDMS chamber with conductive Ag paste to form a 3D electrode; such electrodes were placed on both sides of the microchannel, and cell-separation experiments were performed [26]. Lewpiriyawong et al. used a mixture of Ag powder and PDMS to form a 3D electrode that was placed on one side of the microchannel and separated dead and living yeast cells using dielectrophoresis [27]. Jia et al. designed and fabricated a microfluidic separation device combining 3D electrodes and arch structures to separate Au-plated polystyrene particles and yeast cells via dielectrophoresis [28]. Fu et al. installed a 3D electrode on both sides of the separation channel, the electrode was composed of nanosized carbon black and PDMS, generating an electric field perpendicular to the flow direction that was used for regional electrophoresis and isokinetic electrophoresis [29]. In the studies mentioned above, the composite electrodes were fabricated simply by mixing conductive micro/nanoparticles with PDMS; an in-depth investigation of composite materials for manufacturing 3D microelectrodes has not yet been performed.

This paper introduces a composite 3D microelectrode made of PDMS and Ag nanopowder. The electrical and physical properties of the Ag-PDMS composite were investigated in detail. The electrode had high conductivity, high processability, and good compatibility with a microfluidic chip. Moreover, a processing method for microelectrodes on PDMS microfluidic chips was developed, which can be used to obtain 3D microelectrodes of any shape. A microfluidic chip with a composite 3D electrode was designed and fabricated, and polystyrene particles of three different sizes were successfully separated in experiments.

## 2. Materials and Methods

### 2.1. Mathematical Model for Particle Trajectory

When particles are subjected to an electric field, the positive and negative charges inside the particles are separated, forming an electric dipole. If the electric field is non-uniform, the electric-field strengths on the two sides of the particle differ, and the size of the induced dipole is not equal. The resultant force, which is called the dielectrophoretic force, is nonzero, resulting in particle movement [30]. The time-averaged dielectrophoretic force of a spherical particle in a non-uniform electric field can be expressed as follows:(1)FDEP=2πεma3Re[K(ω)]∇|Erms|2where $F_{DEP}$ represents the dielectrophoretic force to which the particles are subjected, $\varepsilon_{m}$ represents the absolute permittivity of the suspending medium; a represents the particle radius, $\nabla {\left|E_{rms}\right|}^{2}$ represents the gradient of the square of the electric-field intensity, and  K(ω) represents the Clausius–Mossotti factor (also known as the CM factor), which is determined by the electric-field frequency and the complex permittivity of the suspending medium [31].  K(ω) can be expressed as
(2)$$\;K\left(\omega \right)=\frac{{\tilde{\varepsilon }}_{p}-{\tilde{\varepsilon }}_{m}}{{\tilde{\varepsilon }}_{p}+2{\tilde{\varepsilon }}_{m}}$$
where  ω represents the electric-field angular frequency, ${\tilde{\varepsilon }}_{p}$ represents the complex permittivity of the particles, and ${\tilde{\varepsilon }}_{m}$ represents the complex permittivity of the suspending medium. The complex permittivity of an isotropic homogeneous dielectric can be expressed as
(3)$$\tilde{\varepsilon }=\varepsilon -j\frac{\sigma }{\omega }$$
where $j=\sqrt{-1}$, and  ε and  σ represent the permittivity and conductivity, respectively, of the particles or the suspending medium.

As indicated by Equation (2), Re[*K(ω)*] (the real part of *K* (*ω*)) ranges from –0.5 to 1. When ${\tilde{\varepsilon }}_{p}$
*>>*
${\tilde{\varepsilon }}_{m}$, Re[*K(ω)*] tends to 1. When ${\tilde{\varepsilon }}_{p}$
*<<*
${\tilde{\varepsilon }}_{m}$, Re[*K(ω)*] tends to be –0.5 [32]. When Re[*K(ω)*] *>* 0, the particles are subjected to positive dielectrophoresis and move in the direction of increasing electric-field intensity. When Re[*K(ω)*] *<* 0, the particles are subjected to negative dielectrophoresis (nDEP) and move in the direction of decreasing electric-field intensity.

As indicated by Equation (1), the magnitude of the dielectrophoretic force mainly depends on the particle size, electric-field strength, and *CM* factor, and the *CM* factor *K(ω)* is related to the permittivity, conductivity of the solution and particle, and electric-field frequency. The *CM* factor determines whether the dielectrophoretic force is positive or negative, i.e., the direction of particle motion [33]. Importantly, for the particular particles to be sorted, when the corresponding buffer solution is selected, the CM factor is only related to the electric-field frequency.

An electric field intensity is needed to calculate the dielectrophoretic force. Without considering the presence of the particle, the electric potential of the medium is governed by the following equation:(4)$$\nabla \left(\sigma +j\omega \varepsilon_{m}\right)\nabla \varphi =0,$$

An external electric potential bias is applied between the electrodes, and electrical insulation is applied at all the other boundaries.

In the microfluidic chip, the particles are mainly subjected to the dielectrophoretic force and hydrodynamic drag force [34]. For a uniform spherical medium, the drag force acting on the particles can be expressed as follows:(5)$$F_{drag}=6\pi \mu a\left(v_{m}-v_{p}\right)$$
where  μ represents the viscosity of the fluid, a represents the radius of the particle, $v_{m}$ and $v_{p}$ represent the velocities of the fluid and particle, respectively.

To understand and predict the trajectories of particles in our chip, we simulated the electrokinetic motions of spherical particles in a two-dimensional (2D) device model using COMSOL Multiphysics 5.3 (www.comsol.com). A stationary computational fluid dynamics problem was solved by using the physics of creeping flow, and the distribution of the electric field was solved by using the physics of electric current frequency domain. The time-dependent particle trajectories were solved by using the physics of particle tracing for fluid flow. The drag force and dielectrophoretic force were applied to calculate the trajectories of particles. 

The fluid flow was governed by the Stokes equation and the continuity equation:(6)$$-\mu \nabla^{2}u+\nabla p=0$$
(7)∇u=0.
where μ is the viscosity of the fluid, u is the fluid velocity and p is the fluid pressure.

Different flow rates were applied at the inlet, and a normal flow with no external pressure gradient (i.e., *p* = 0) was set at the outlet. The no-slip boundary condition was applied at the channel walls.

The translation of the particle was given as follows:(8)$$F=m\frac{dv_{p}}{dt}=F_{DEP}+F_{drag}$$

Two techniques were used. The first solves for the steady-state fluid dynamics and frequency-domain (alternating current (AC)) electric potential, and the second uses a time-dependent study step that utilizes the solution from first method to estimate the particle trajectories.

### 2.2. Experiment

#### 2.2.1. Dispersion of Ag Powder

Smaller Ag powder particles are more likely to spontaneously gather because of the electrostatic force [35]. An appropriate method for controlling the dispersion of particles in the composite matrix is essential for achieving the required material properties. In this study, direct and indirect dispersion methods were used to prepare composite materials, and the electrical properties and hardness of the composite materials were compared.

In direct dispersion, Ag nanopowder and PDMS (Dow Corning, Sylgard 184, Midland, MI, USA) were poured into a cup. A high-speed disperser with 2000 r/min was used to transform larger agglomerates into smaller ones, under a large shearing force. This method could sufficiently disperse fine Ag particles in the PDMS matrix. However, it is more effective for larger agglomerates and thus has limitations.

In indirect dispersion, ethyl acetate (EAC) which is a low-viscosity liquid was used as an auxiliary dispersion solvent to dissolve PDMS prepolymer and curing agent. First, Ag nanopowder (Xfnano, Nanjing, China) was dispersed into EAC (AR, Chongqing, China) using ultrasonication in a cup. Second, PDMS prepolymer was dissolved into EAC in the other cup, the viscosity of PDMS was significantly reduced. Third, mixed the liquid in the two cups using stirring and ultrasonic dispersion. The EAC was evaporated in 50 °C for 24 h. PDMS curing agent was added into the mixture of Ag nanopowder and PDMS prepolymer using high-speed disperser with 2000 r/min for 1 h. Finally, a uniformly dispersed composite material was obtained.

#### 2.2.2. Performance Evaluation of Composite Material

To accurately test the conductivity of the Ag-PDMS composite material, we designed and fabricated a rectangular parallelepiped composite sample with the same geometry. The length, width, and height of the rectangular parallelepiped were 10, 5, and 5 mm, respectively. Spherical Ag powder with particle sizes of 10 nm, 100 nm, and 1 μm was selected as the conductive filler for the composite. First, the correlation between the proportion of the filler corresponding to the three particle sizes of Ag powder and the electrical resistance was investigated. The resistance of low-filler ratio composites with a high resistance was measured using a megohmmeter. The resistance of high-filler ratio composites was tested using a high-accuracy balanced bridge. Additionally, the hardness of the composite material was considered, as it affected the subsequent process. A shore scale hardness tester (Wallace Instruments) was used to evaluate the mechanical properties of the composite materials.

The resistance and hardness measurement results are shown in Figure 1A and Figure 1B, respectively. When the proportion of Ag powder (mass fraction) was <60% *w*/*w*, the composite exhibited a high resistance (up to several hundred megohms), low hardness, and good mechanical properties. When the proportion increased to 70% *w*/*w*, the electrical resistance of the composite decreased exponentially; the material had better electrical conductivity, with increased hardness and poor mechanical properties. When the proportion reached 75% *w*/*w*, the electrical resistance of the composite was several hundred ohms, and the composite had good electrical conductivity, with high hardness and low flexibility. When the proportion of Ag powder was >80% *w*/*w*, although the electrical resistance of the composite material was several tens of ohms, owing to the low PDMS content and excessive Ag powder, the entire composite was not firmly bonded, causing the Ag powder to easily fall from the composite. Additionally, the hardness of the composite material was very high, resulting in high brittleness and poor mechanical properties. For composite materials with the same proportion of Ag powder, a smaller particle size corresponded to a lower resistance and lower hardness. The resistance was lower with EAC than without EAC, and the composite with EAC exhibited a lower hardness and better mechanical properties, because the PDMS prepolymer was dissolved in the EAC, reducing the viscosity and allowing the Ag powder to be more uniformly dispersed in the composite.

In order to research the effect of Ag nanoparticles concentration on the electrical capacitance of the composite materials, 10-nm Ag powder and PDMS were mixed using indirect dispersion. The AC electrical resistance was measured in the frequency range of 0–10^8^ Hz, and the results are shown in Figure 1C. With the increase of frequency, the resistance of composite electrode (60% and 70% *w*/*w*) decreases continuously. Conductive network and a lot of small capacitance composed of PDMS and Ag nanopowder coexist in the composite electrode. The electrical conductivity of composite materials (40% *w*/*w*) was very bad because the conductive network has not yet been formed. The resistance of composite materials (75% *w*/*w*) was very small, the electrons move along the dense conductive network.

When the mass ratio of silver powder is higher than 75% *w*/*w*, the resistance of composite electrode does not decrease significantly with increase of Ag nanopowder, but the hardness increases continuously, as shown in Figure 1A,B. We found that the hardness of composite material was larger than 60 SH.A. The difficulty of microfabricating the 3D electrodes increased remarkably in the experiment. The experimental results indicated an inverse correlation between the conductivity and the hardness of the composite materials; with the increasing concentration of Ag nanopowder, the conductivity improved, and the hardness deteriorated. Considering the conductivity and the adaptability with a microfluidic chip, 10-nm Ag powder was used as the conductive filler, a mass ratio of 75% *w*/*w* was used as the optimum equilibrium point, EAC was added as a mixed solvent, and the resistivity and hardness of the composite were 0.26 Ω·m and 59.2 SH.A, respectively. The distribution of Ag powder inside the composite material was examined via SEM, as shown in Figure 1D. The SEM images indicated that the spherical Ag powder particles were firmly wrapped by PDMS and that the nanoparticles were in contact with each other, forming a conductive network.

#### 2.2.3. Microfabrication

Microfluidic chips for the separation of polystyrene particles were manufactured using soft lithography techniques [36], as shown in Figure 2A. To obtain a 3D microelectrode, a groove with a width of 200 μm and a height of 50 μm was fabricated on a 3-inch Si wafer (ePAK, Texas, USA) through the SU-8 3050 (Microchem, Massachusetts, MA, USA) negative photoresist. The groove was filled with the Ag-PDMS nanocomposite obtained in the previous step, and the excess material was scraped off using a sharp and flat blade. Then, the Si wafer was baked in an oven for 1 h at a temperature of 85 °C, and PDMS (Dow Corning, Midland, MI, USA) was poured onto the Si wafer. Subsequently, baking was performed again at 85 °C for 1 h for the adhesion of the composite electrode, yielding a PDMS substrate with a composite 3D microelectrode [37]. Using the same procedure, another substrate with a microchannel was fabricated. The microchannel had a width of 200 µm and a height of 50 µm, and the chip was perforated at the two inlets and three outlets. After a 2-minute pretreatment using an O_2_ plasma cleaner (PDC-MG, Chengdu, China) to hydroxylate the surface, the substrate with the 3D microelectrode and the substrate with the microchannel were bonded using a microalignment platform [38]. Finally, after baking on a hotplate at 150 °C for 10 min, a microfluidic chip with 3D sidewall electrode and a closed channel structure was fabricated, as shown in Figure 2B. The 3D microelectrode was embedded into the groove of the electrode, and the head of the electrode was flush with sidewall of the microchannel, as shown in Figure 2C.

#### 2.2.4. Experimental Solutions

In this study, polystyrene particles were considered as the objects to be separated. The samples were mixed with polystyrene particles (BaseLine, Tianjin, China) of different sizes: 20, 10, and 5 μm. The concentration was adjusted to 10^3^−10^4^ particles per milliliter using phosphate-buffered saline (PBS, Solarbio, Beijing, China). To prevent adhesion among the particles and between the particles and the chip, Tween 20 (TP1379, Bomeibio, Hefei, China; concentration of 0.1% *v*/*v*) was added to the buffer solution [39]. The concentration of the PBS buffer used in the experiment was 1-mM, and the electrical conductivity was 0.17 S/m. Reducing the conductivity can, to a certain extent, reduce the Joule heating effect produced in the experiment and its effect on the activity of the sorted sample [40].

#### 2.2.5. Experimental Manipulation and Visualization

To observe the particle separation process under the electrokinetic force, an experimental platform was established, which consisted of a function signal generator (SDG1020, Shenzhen, China), a voltage amplifier (ATA-2042, Agitek, Xian, China), an injection pump (LSP02-2B, Longer, Baoding, China), a microfluidic chip, an inverted fluorescence microscope (IX73, Olympus, Tokyo, Japan), and a digital camera (MSX2, Mshot, Guangzhou, China).

Before the experiment, the microfluidic device was washed with 1-mM PBS buffer without particles for 5 min. The chip inlet was connected to the syringe on the injection pump by a catheter. The injection pump drove the syringe to inject liquid into the two inlets of the chip. Inlet 1 was filled with PBS buffer (concentration of 10%) at an injection rate of 3 μL/min, and inlet 2 was injected with a suspension containing polystyrene particles of three sizes at an injection rate of 1 μL/min. The signal produced by the generator was amplified by the voltage amplifier and applied to the composite electrode through a flat port clamp. The experimental phenomena were observed using an inverted microscope and recorded by a digital camera. 

## 3. Results and Discussion

### 3.1. Numerical Simulation

Positive and negative values of the real part of the CM factor indicated that the particle was subjected to a positive and negative dielectrophoretic force in the electric field, respectively. The magnitude mainly depended on the characteristic parameters of the particle, solution, and electric-field signals. In the experiment, a PBS buffer with a concentration of 1 mM was used, and its electrical conductivity σ_m_ and dielectric constant ε_m_ were 0.17 S/m and 7.04 × 10^−10^ F/m, respectively. The conductivity of the polystyrene particles was calculated using the formula σ_p_ = 2K_s_/a, where a represents the radius of the polystyrene particles, and K_s_ = 10^−9^ S represents the surface conductance of the polystyrene particles. The dielectric constant ε_p_ of the particles was 2.04 × 10^−10^ F/m. The real part of the CM factor was calculated using Equations (1)–(3). The calculation results indicated that when the frequency of the signal (ω) was 0–10^7^ Hz, the real part of the CM factor was always negative. Therefore, the polystyrene particles were constantly subjected to the negative dielectrophoretic force in this frequency range. Additionally, we analyzed the electrolytic effects of the chips at different frequencies. The microelectrode was more susceptible to electrolysis at a low frequency than at a high frequency. To ensure that the electrode was not electrolyzed, the signal frequency of the electric field was >10 KHz. Considering Re[K(ω)] and the electrode electrolysis, the electric-field frequency was selected as 1 MHz in this study. At this frequency, the particles were subjected to negative dielectrophoretic forces, and the electrodes were not electrolyzed.

To optimize the injection speed and the amplitude of the applied signal for efficient separation of polystyrene particles with different sizes, a 2D separated model was developed in COMSOL Multiphysics 5.3 (COMSOL, Newton, MA, USA). The model was used to analyze the distribution of the flow field, the electric-field gradient, and the electric-field intensity inside the separation chip, which affected the particle movement trajectory and separation efficiency in the microfluidic chip. Figure 3A shows the flow-field distribution inside the chip.

When the chip injection flow path is Y-shaped, low-concentration PBS buffer flows into both ends, which can be regarded as a Newtonian liquid. Owing to the incompressible and laminar liquid, the two flows of inlet 1 and 2 are still laminar at the time of meeting, and no mixing liquid is created. Instead, an interface is formed between the two fluids, which occupy widths of $d_{1}$ and $d_{2}$. Because the flow is incompressible, the total amount of influent liquid is equal to the total amount of outflow, and the width of the two liquids is proportional to their volume:(9)$$\frac{d_{1}}{d_{2}}=\frac{v_{1}}{v_{2}}=\frac{dp_{1}}{dx}/\frac{dp_{2}}{dx}$$

Equation (9) indicates that the width of the liquid is related to its speed ($v_{1}$ and $v_{2}$) or the pressure ($p_{1}$ and $p_{2}$) at the end of the injection. If the injection speed of the two inlets or the pressure of the injection end is known, the widths of the two fluids in the main channel can be calculated [41].

Because the two samples (Newtonian fluids) satisfied the laminar flow conditions of Poiseuille’s law, they combined at the Y-type structures when the injection-speed ratio of inlet 1 to inlet 2 was 3:1, according to Equation (9), the width ratio of flow of inlet 1 and inlet 2 was also 3:1. Therefore, the polystyrene particles were compressed to flow toward the chip outlet on the side close to the electrode.

The gradient of the square of the electric-field intensity in the microfluidic channel is shown in Figure 3B. The simulation results indicated that the gradient of the electric field was relatively large in the area adjacent to the composite electrodes and was largest at the corners of the electrodes. When a signal (Vpp: 20 V; frequency: 1 MHz) was applied, the maximum gradient of the square of the electric-field strength in the separation channel having a width of 200 μm was 3.72 × 10^16^ V^2^/m^3^. 

To demonstrate the advantages of 3D comparing planar electrodes, a 3D model was carried out by COMSOL. A line on the vertical section in the middle of the separation channel was selected to research distribution of the electric-field intensity, as shown in Figure 3C,D. The electric-field intensity of this line from the bottom (Y = 0-μm) to top (Y = 50-μm) is shown in Figure 3E,F. When the same signal (Vpp: 20 V; frequency: 1 MHz) was applied at the same position, the electric-field intensity of 3D sidewall electrode was more than twice that of base electrode, and the electric-field intensity of the 3D electrode remains stable in vertical direction, while the base electrode changes significantly.

Under the negative dielectrophoretic force, particles were pushed away from the electrode. Because *F_DEP_* was proportional to the cube of the particle radius a, the lateral displacement of the particles was proportional to the cube of the particle radius. Finally, the three types of particles flowed out of three different outlets, as desired. The results of numerical simulation show the motion trajectory of particles with three different sizes under the different dielectrophoretic force and fluid driving. 

### 3.2. Effect of Voltage on Separation

Because the separation throughput depended on the flow rate, *v*_1_ = 3 μL/min and *v*_2_ = 1 μL/min were selected as the injection flow rates. Under these flow conditions, the separation efficiency significantly depended on the amplitude of the applied signal, because the signal amplitude directly affected the dielectrophoretic force and the longitudinal displacement of the particles. Before the experiment, the effects of the signal voltage on the separation efficiency were numerically simulated using the developed model. The results indicated that the applied signal affected the separation process.

When the amplitude of the applied signal was <10 V, the three types of particles could not be separated by the negative dielectrophoretic force. For 10 V < Vpp < 15 V, the 20-μm particles had a large longitudinal migration because of the strong dielectrophoretic force. However, the applied signal could not induce a sufficient force to drive the 20-μm particles to move to outlet 1 and finally enter outlet 2. The 10-μm polystyrene particles had insufficient longitudinal migration to enter outlet 3 with the 5-μm polystyrene particles, as shown in Figure 4(A1). When the applied signal amplitude was in the range of 15 V < Vpp < 19 V, the longitudinal migration of the 20-μm polystyrene particles was still insufficient, and they entered outlet 2. The longitudinal migration of the 10-μm particles subjected to the dielectrophoretic force increased, and then the particles entered outlet 2, while the 5-μm particles still entered outlet 3, as shown in Figure 4(A2). When the signal amplitude was increased to 19 V < Vpp < 25 V, the 20- and 10-μm polystyrene particles had sufficient and suitable longitudinal migration, entering outlets 1 and 2, respectively, under the negative dielectrophoretic force. The 5-μm polystyrene particles exhibited a small longitudinal migration but could not enter outlet 2 and continued to flow out of outlet 3, as shown in Figure 4(A3). These results satisfy the requirements of separation and support the selection of 19–25 V as the optimum amplitude range of the applied signal. For 25 V < Vpp < 30 V, the 5-μm polystyrene particles were considerably shifted in the longitudinal direction and entered outlet 2 together with the 10-μm polystyrene particles, and the 20-μm polystyrene particles continued to flow out of outlet 1, as shown in Figure 4(A4). For 30 V < Vpp < 40 V, the 10-μm polystyrene particles were excessively shifted in the longitudinal direction and flowed out of outlet 1 together with the 20-μm polystyrene particles, and the 5-μm polystyrene particles were excessively shifted in the longitudinal direction and flowed out of outlet 2, as shown in Figure 4(A5).

### 3.3. Effect of Flow Rate on Separation

The flow rate played a decisive role in the control of the particle movement in the microfluidic channel, with significant effects on the separation throughput, accuracy, and efficiency. To fix peak-to-peak voltage of separation signal, we adjusted the injection flow rates of inlets 1 and 2 and examined the effects on the particle separation process. Similar to the analysis of the voltage effect on the separation, the separation process was numerically simulated under different flow rates using COMSOL before experimental research. Expected results for the separation at different flow rates and the flow-rate ratio of inlet 1 to inlet 2 were obtained, as shown in Figure 4(B1–B5).

The experimental results indicated that particles in the starting region of the microfluidic channel were compressed close to the sidewall with composite 3D microelectrodes under the injection flow of inlets 1 and 2. Then, these particles were pushed away from the electrode under the negative dielectrophoretic force (Vpp:20V) and flowed to the outlet together with the fluid. When the injection flow rate increased, the separation time (the duration of particle flow in the separation microfluidic channel) decreased. In this case, the longitudinal migration was too small for the particles to move to the desired outlet, as shown in Figure 4(B1). At *v*_1_ = 6 μL/min and *v*_2_ = 2 μL/min, the longitudinal migration of the 20-μm polystyrene particles was insufficient; thus, the particles flowed to the wrong exit (outlet 2 instead of outlet 1). When the injection flow rate was low, the polystyrene particles needed a long time to pass through the microfluidic channel, and the longitudinal migration of the large particles was sufficient for accurate sorting, as shown in Figure 4(B5). Under the flow-rate conditions of *v*_1_ = 1.5 μL/min and *v*_2_ = 0.5 μL/min, the longitudinal migration of the 5-μm polystyrene particles was extremely large; consequently, the particles entered the wrong exit (outlet 2 instead of outlet 3).

When the injection flow rates of inlets 1 and 2 and the flow-rate ratio were changed, the positions of the particles that were compressed on the side close to the electrode were changed when the particles entered the separated channel from inlet 2. According to Equation (9), at *v*_1_ = 6 μL/min and *v*_2_ = 1 μL/min (*v*_1_:*v*_2_ = 6:1), the sample was injected from the two inlets, and the liquid width ratio was *d*_1_:*d*_2_ = 6:1. Under this condition, the particles were compressed into the electrode with a width of approximately 30 μm. The particles easily entered the liquid layer of inlet 1 under the negative dielectrophoretic force. Then, the flow rate of the particles moving to the outlet suddenly increased, and the particles (large and small) flowed to the outlet rapidly without sufficient longitudinal migration. The 20- and 10-μm particles flowed into outlets 2 and 3, respectively, as shown in Figure 4(B2). At *v*_1_ = 1.5 μL/min and *v*_2_ = 1 μL/min (*v*_1_:*v*_2_ = 1.5:1), the liquid width ratio was *d_1_:d_2_* = 1.5:1 at the convergence of the two inlets. Because of the long distance from the electrode and the low flow rate, the mixed particles remained in the separation channel for a long time and had an excessive longitudinal migration. The 5-μm particles with an excessive longitudinal migration flowed to the wrong exit (outlet 2 instead of outlet 3), as shown in Figure 4(B4). At *v*_1_ = 3 μL/min and *v*_2_ = 1 μL/min (*v*_1_:*v*_2_ = 3:1), the particles of different sizes had the appropriate longitudinal migration and flowed to the desired outlets, as shown in Figure 4(B3). At *v*_1_ = 3 μL/min and *v*_2_ = 1 μL/min, the flow rate and negative dielectrophoretic force achieved a relatively stable equilibrium, resulting in the appropriate offset of the 20-, 10-, and 5-μm particles and ensuring that they flowed into outlets 1, 2, and 3, respectively.

### 3.4. Separation of Three Different Polystyrene Particles

According to the numerical simulation, the injection flow rate and separation voltage were optimized to achieve continuous separation with a high efficiency and accuracy. A 19–25 Vpp/1 MHz applied signal and injection flow rates of *v*_1_ = 3 μL/min and *v*_2_ = 1 μL/min were appropriate for the separation of 20-, 10-, and 5-μm particles using the microfluidic device. Under these conditions, the polystyrene particles of three different sizes quickly entered the appropriate outlets. Via classification and statistical analysis of the experimental results, the separation efficiency of the Ag-PDMS composite 3D microelectrode was calculated. Figure 5(A) shows the separation process of 20-, 10-, and 5-μm polystyrene particles under a 21 V/1 MHz AC electric signal and injection rates of *v*_1_ = 3 μL/min and *v*_2_ = 1 μL/min. The particle throughput of this device was approximately 50/min. High-purity particles were collected at each outlet, as shown in Figure 5(B1–B3). The sorting accuracy for the 20-, 10-, and 5-μm particles was >90%, as shown in Figure 5(C1–C3).

## 4. Conclusions

A composite material of 10-nm Ag powder and PDMS was introduced, and the effects of the content of Ag nanoparticles on the electrical conductivity and hardness of the composites were investigated. An optimized Ag-nanopowder dispersion method in which volatilizable EAC is used as a dispersing agent was developed. To balance the conductivity and hardness of the composite material, 75% *w*/*w* of 10-nm Ag powder was selected for the Ag-PDMS composite. Because of the good plasticity and flexibility of the composite material, a fast, easy, and low-cost micromachining process was designed for fabricating 3D microelectrodes of any shape. Using the microfluidic chip with a composite 3D electrode, polystyrene particles of three different sizes were successfully separated by adjusting the flow rate and dielectrophoretic force. The experimental results indicated that continuous separation of polystyrene particles with a high throughput, accuracy, and efficiency microfluidic chip system that had the advantages of no contact, was achieved using the microfluidic device, and high-purity particles (>90%) were collected. The simple operation, and no effect on the biological activity of the sample, are beneficial for reusing and reculturing after separation. 

The chip with Ag-PDMS composite 3D microelectrodes has broad application prospects in the field of microfluidics, e.g., the separation of different types of tumor cells and the development of on-chip Ag/AgCl reference electrodes for measuring the potential and pH in chip channels [42]. Additionally, owing to the flexibility of Ag-PDMS, the chlorinated composite material can be used to fabricate flexible electrodes for wearable devices [43] for detecting EEG or EMG signals without polarization phenomena.

## Figures and Tables

**Figure 1 micromachines-11-00700-f001:**
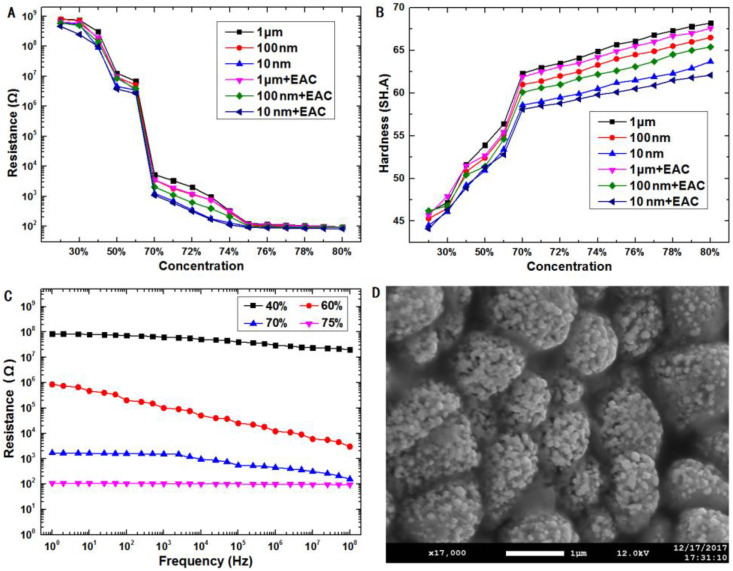
Effects of the Ag powder particle size and dispersion method on the (**A**) resistance and (**B**) hardness of the composite. (**C**) Alternating current (AC) electrical resistance of composite electrode with different mass ratio, mixed 10-nm Ag powder and polydimethylsiloxane (PDMS) using indirect dispersion. (**D**) Scanning electron microscopy (SEM) images of composite electrode (75% *w*/*w*), mixed 10-nm Ag powder and PDMS using indirect dispersion.

**Figure 2 micromachines-11-00700-f002:**
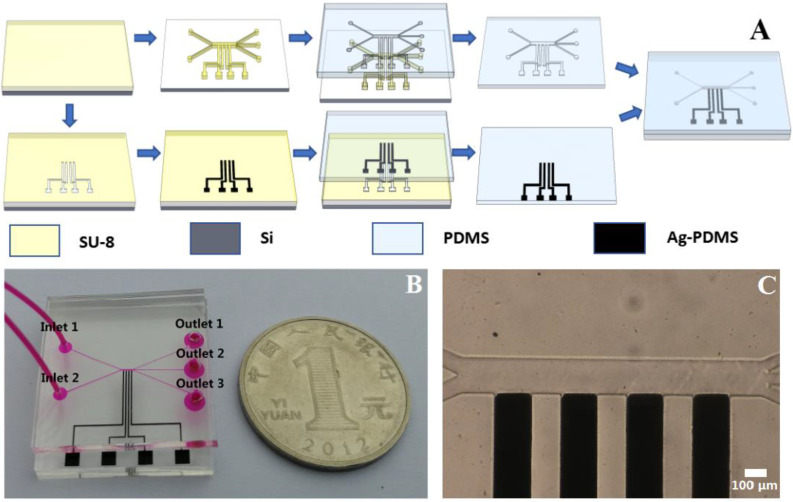
(**A**) Micromachining flowchart for fabricating a microfluidic device with 3D electrodes. (**B**) Micrograph of the microfluidic chip with the composite 3D electrode. The channel and electrode widths were both 200 μm. (**C**) Photograph of a microfluidic chip.

**Figure 3 micromachines-11-00700-f003:**
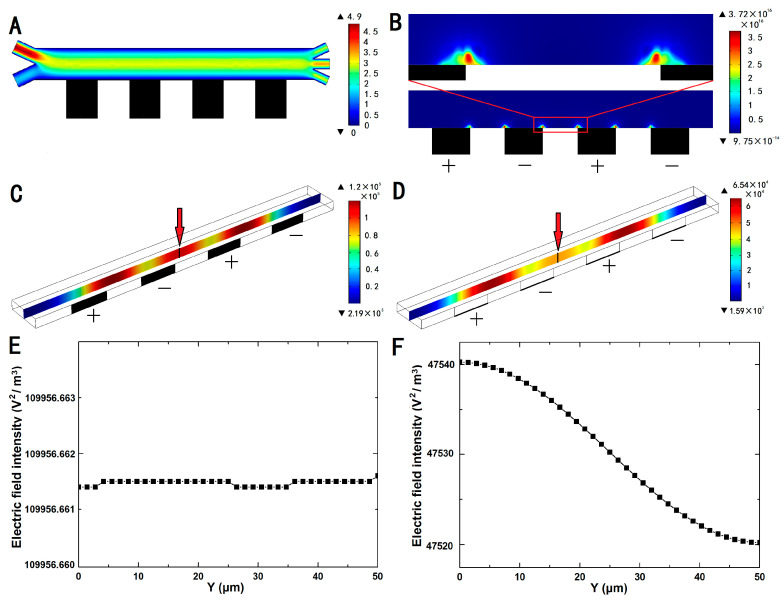
(**A**) Flow-field simulation of the separation process. The flow rates of inlets 1 and 2 are *v*_1_ = 3 μL/min and *v*_2_ = 1 μL/min, respectively. (**B**) Gradient of $\nabla {\left|E_{rms}\right|}^{2}$ near the electrode (Vpp = 20 V; the electric-field frequency was 1 MHz). Distribution of the electric-field intensity on the vertical section of separation channel, (**C**) 3D and (**D**) planar electrodes. The electric-field intensity of the line in the middle of vertical section from the bottom (Y = 0-μm) to top (Y = 50-μm), (**E**) 3D and (**F**) planar electrodes.

**Figure 4 micromachines-11-00700-f004:**
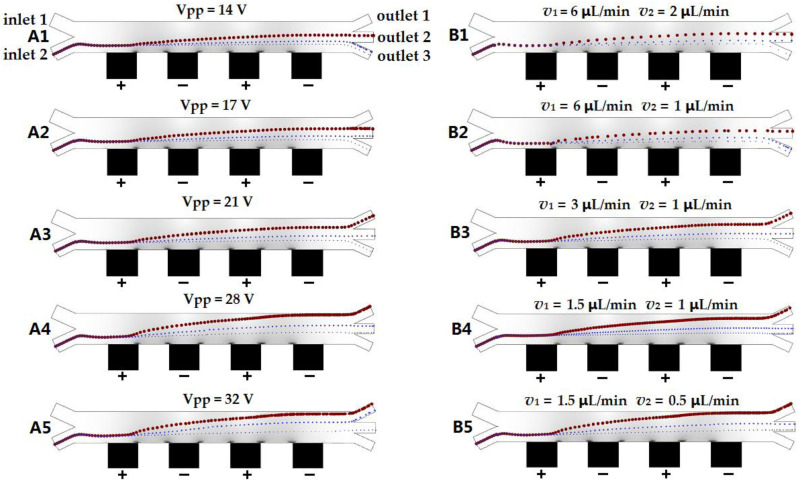
(**A1**–**A5**) Numerical simulation for the effect of changing the peak-to-peak voltage (Vpp) of the separation signal on the separation results at fixed flow rates of *v*_1_ = 3 μL/min and *v*_2_ = 1 μL/min. (**B1**–**B5**) Numerical simulation for the effects of the flow rate and the ratio of the flow rates at the two inlets on the separation at a fixed peak-to-peak voltage of 20 V.

**Figure 5 micromachines-11-00700-f005:**
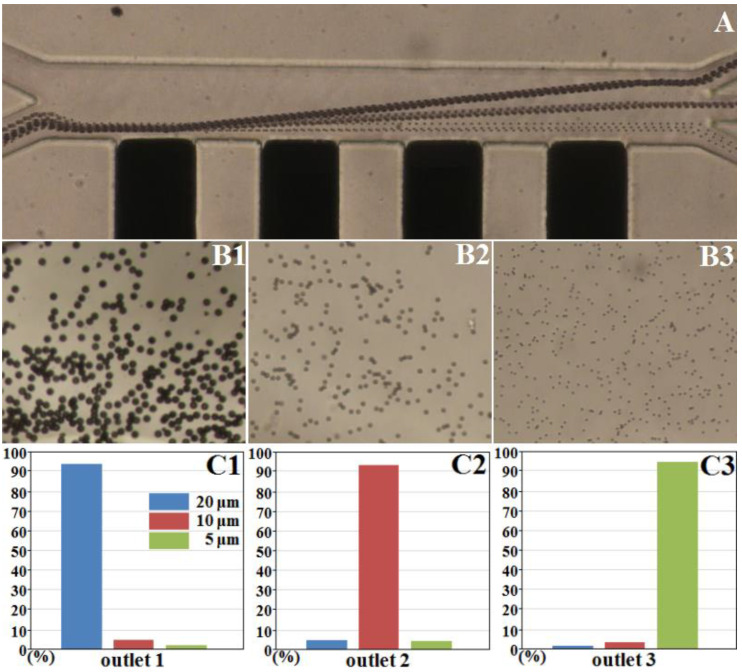
(**A**) Continuous separation of polystyrene particles of three sizes via nDEP. (**B1**–**B3**) Separated particles at the chambers of outlets 1, 2, and 3, respectively. (**C1**–**C3**) Histograms indicating the percentages of the three types of particles in each outlet.

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
