# Peer review of "Dielectrophoretic Separation of Particles Using Microfluidic Chip with Composite Three-Dimensional Electrode"

_micromachines, 2020, doi:10.3390/mi11070700_

Round 1

Reviewer 1 Report

In this manuscript, the authors report design of '3D' composite-electrodes for separation particles, with a balance between conductivity and softness for future applications of flexible electronic devices. However, the research work can be made more comprehensive; and the novelty is limited as many researchers have already reported similar works in the past. The authors should address the following concerns below.

- The ‘3D’ structure of the electrodes has not been shown to induce any improvements on the cell separation applications, as there is no comparison with the ‘2D’ case. There is a significant thickness of the electrodes but it seems that the sidewalls are still close to planar surfaces.

- In Figure 1a and 1b, how can the smooth lines in the plots be obtained from the measurements?

- In section 2.2.1, indirect dispersion involves Ag powder (First step) and Ag nano-particles (Third step) into PDMS. Can you provide the detailed quantity of Ag in each of these steps?

- Details for fabrication 3D electrodes are not clearly described in section 2.2.1.

- Analysis and experimental characterization on the ratio between Ag and PDMS for the good balance between conductivity and softness is not included.

- Besides, the possibility changes in material properties (i.e. conductivity) under physical deformation of the device substrate are not discussed and investigated.

- If there is no plan for including cell/bio-related separation experiment in the revision later, please remove all the statements about applications on cell and biological samples in the manuscript.

Reviewer 2 Report

In this study, Chen et al. have carefully studied Ag-PDMS composite materials for manufacturing 3D microelectrodes, presented the fabrication process, and demonstrated separation of 20, 10 and 5 µm polystyrene particles. The results presented in this work will benefit researchers using Ag-PDMS microelectrodes in the electrokinetics field. Manuscript is well written, and I recommend publication subject to minor revision:

1. Authors have analyzed the effects of the content of Ag nanoparticles on the electrical conductivity and hardness of the composites. I would strongly recommend authors to also analyze two additional important parameters

a. Maximum de-lamination pressure of the bonded PDMS devices. Does the concentration of Ag nanoparticles affect the maximum allowable pressure that a PDMS device consisting of these microelectrodes can sustain? Do we have an optimum concentration where a device can sustain 50 to 80 Psi pressures that are typical of the standard PDMS devices?

b. Effect of Ag nanoparticles concentration on the electrical capacitance of the composite structure

2. Authors should provide more details on the COMSOL computation model. It is an important part of the paper. Did they perform mesh independence study?

There are a few places where grammatical errors are present, for example line 62 on page 2: “However, pure PDMS cannot used as an electrode” should be “However, pure PDMS cannot be used as an electrode.” I would recommend to double check for those errors.

Reviewer 3 Report

please see attached

Round 2

Reviewer 1 Report

Thank you for the quick response of the authors. To avoid any possible misunderstanding of the readers, once again, the term ‘3D’ in this work sounds a bit misleading. The electrodes developed in the work have a certain thickness, yet the functional vertical surfaces are still ‘planar’. It is suggested to change the terms ‘2D’ and ‘3D’ throughout the paper. For instance, ‘2D electrode’ may be changed to ‘base electrode’ and ‘3D electrode’ may be changed to ‘sidewall electrode’.